# A 26.2:1 Bandwidth Ultra-Wideband Low-Profile Tightly Coupled Dipole Array with Integrated Feed Network

**DOI:** 10.3390/s25113418

**Published:** 2025-05-29

**Authors:** Bailin Deng, Yu Yang, Xiuyuan Xu, Eryan Yan, Hongbin Chen

**Affiliations:** 1Institute of Applied Electronics, China Academy of Engineering Physics, Mianyang 621907, China; dengbailin22@gscaep.ac.cn (B.D.); yangyu1004@126.com (Y.Y.); xuxiuyuan22@gscaep.ac.cn (X.X.); yaneryan_2002@163.com (E.Y.); 2Graduate School of China Academy of Engineering Physics, Beijing 100193, China

**Keywords:** ultra-wideband antennas, frequency selective surfaces (FSS), dipole antennas

## Abstract

**Highlights:**

**What are the main findings?**

New structure dipole antenna.Minimalist feeding method.

**What is the implication of the main finding?**

Dipoles close to each other in an arc shape between elements, making the mutual coupling and bandwidth increase greatly.The specially designed feed network makes the feed mode greatly simplified and the profile height extremely low.

**Abstract:**

This article presents a novel tightly coupled dipole array (TCDA) with a bandwidth of 26.2:1 (VSWR < 3) across 0.20–5.23 GHz. By adding a new dual-stopband resistive frequency selective surface (RFSS) between the dipole and the floor, the short-circuit points formed by the floor at the frequency points corresponding to *λ* = 2 h and h are both eliminated (h is the height from the antenna to the floor). A specially integrated feed network is also applied to significantly reduce the complexity and profile height to 0.05 *λ_low_*. The simulation and experimental results show that the designed TCDA has extremely wide bandwidth, good directivity and beam scanning potential. Compared with previous designs, it greatly extends the bandwidth, improves the gain, reduces the profile height, and simplifies the feeding method.

## 1. Introduction

With the increasing demand for transmission rate and distance in civil and military fields such as mobile communication, remote control, data transmission, and radar, antennas with larger bandwidth and higher gain have gradually become a hot research and development trend [1]. Ultra-wideband array antennas satisfy this demand with their ultra-wide operating bandwidth, great scanning characteristics and high gain, which have broad application prospects. Among them, the tightly coupled dipole array (TCDA) has received extensive attention from researchers in recent years. Different from the traditional design idea, TCDA makes use of the coupling effect between the units to further extend the bandwidth of the antenna, which has many advantages such as small size, ultra-wideband, low profile, wide angle scanning, low cost, easy processing and conformal. They make TCDA particularly suitable for platforms like UAVs that require strict size constraints and wide operational bandwidth.

The design idea is derived from an ideal infinite continuous current sheet array proposed by Wheeler [2]. Subsequently, Munk found that if the tightly arranged short dipoles’ ends are introduced into the coupling capacitance component, it can exhibit ultra-wideband characteristics [3]. Based on this theory, he designed the TCDA which confirms the feasibility of the above hypothesis [4]. Then, Moulder added a superstrate and a resistive frequency selective surface (RFSS) for TCDA, making its bandwidth reach 21:1 (0.28–5.91 GHz) [5].

However, the RFSS in previous designs can only absorb electromagnetic waves in a single frequency band, which cannot deal with the high-order reflected wave of the floor [5,6]. The existing multi-stopband RFSS uses a multi-layer or staggered serrated structure to absorb high-order waves [7,8,9,10]. They all have complex structures, which will bring higher processing difficulty and cost. Meanwhile, the wide-angle impedance matching (WAIM) is too thick with a height of about 10 mm [11]. More importantly, each unit requires an independent feed port [1,5,8,9,10,11,12,13,14], which makes the feed method of the entire TCDA more complicated and generates additional profile height (as shown in Figure A1). Even for TCDA designed specifically for car platforms, its profile height can only be controlled at 0.087 *λ_low_* [15]. All these issues make it unsuitable for applications like drones that have strict size requirements.

Addressing these challenges, this paper proposes a simplified dual-stopband RFSS and uses it to eliminate two short-circuit points of the floor. Moreover, high-dielectric constant WAIM is used to reduce the profile height and extend bandwidth. An integrated feed network is also designed to decrease the number of feed ports and further reduce the profile height. Eventually, the designed TCDA reached an ultra-wideband bandwidth of 26.2:1 (VSWR < 3 from 0.20 to 5.23 GHz), 0.05 *λ_low_* profile height and only two feed ports.

In Section 2, the TCDA unit cell and RFSS are introduced. In Section 3, TCDA elements are constructed into an array. To feed it, a feed network is designed on the underside of the ground. In Section 4, we manufacture and test TCDA to verify the feasibility of the simulation.

## 2. Unit Cell and RFSS Design

Figure 1 shows the proposed TCDA element, which is constructed and analyzed as infinite periodic units in simulation software HFSS 2023R1. The uppermost layer is the WAIM used to improve the scanning characteristics of the array. Kappa438 with *ε_r_* of 4.38 is selected as the material of WAIM because its thickness is negatively correlated with the dielectric constant. As a consequence, the thickness *h_1_* is lowered from more than ten millimeters in the previous work to 2.5 mm, meaning the height and weight of the element are reduced.

Below the WAIM is a novel dipole printed on RO4003. Its biggest feature is that an inner-cladding structure is used to form a gap close to a circle. In previous works, the dipoles of two adjacent units are usually printed alternately on the upper and lower sides of the substrate. The coupling is carried out by a linear gap formed by overlapping parts on the vertical plane. (shown in Figure A2) [5,6,7,8,11,12,13]. In addition, there are some other coupling structures, but their coupling is still limited [16,17,18,19,20,21].

Different from them, this paper proposes a new dipole structure and coupling method. All dipoles are printed on the upper side of the substrate. The two ends of the adjacent units are, respectively, designed as circles and rings wrapped around them. The gap spacing between adjacent units is only 0.1 mm. Such a design greatly increases the coupling effect between adjacent units by generating longer gap lengths and smaller gap spacing to achieve wider bandwidth. RFSS is placed between the antenna and the ground to absorb the reflected wave at some frequency points generated by the ground and avoid its inverse superposition with the upward electromagnetic wave. The dipole is fed by a twin coaxial line. It is a structure composed of two inner conductors and an outer conductor which connected to the dipole and the floor, respectively. This enables 180° differential feeding with small loss and easy adjustment. The height *h* = 71.4 mm from the antenna to the floor is set to half of the wavelength *λ_mid_*, which corresponds to the original center frequency *f_mid_* = 2.1 GHz.

In previous works, single stopband RFSS is usually used to absorb the reflected wave of *f_mid_*. This is because the electromagnetic wave radiated downwards by the antenna at *f_mid_* will generate a phase difference of 180° with the upward wave, resulting in an equivalent short-circuit point. However, it has high orders at the integer times of *f_mid_*, which prevents the bandwidth from expanding to high frequency.

To solve this problem, we present a novel dual-frequency RFSS with a dual-ring structure. Its working principle is to use two rings of different sizes to resonate with frequency points of different wavelengths to prevent electromagnetic waves from passing through. By adjusting the side length, width, resistivity of the two resistance rings and their height from the ground, a great frequency selection characteristic is obtained. The resistance values of the inner and outer rings are 52.5 Ω and 76.1 Ω, respectively.

Compared with multilayer structures or serrated structures, this design uses the simplest structure that can exist in theory to achieve multi-stopband function. This significantly reduces design difficulty, processing difficulty, and costs, and has significant application advantages.

Unlike metal frequency selective surface, RFSS will generate current on its surface due to the existence of resistance value. The energy of electromagnetic waves is converted into heat and distributed rather than reflected back, which is more suitable for TCDA applications with floors. As shown in Figure 2, designed RFSS can absorb electromagnetic waves of *f_mid_* and 2*f_mid_* at the same time.

Active VSWR of the final TCDA is depicted in Figure 3. Compared with the case without RFSS, TCDA with new RFSS can extend the bandwidth to 5.2 GHz, which contains 2.4 and 5 GHz common communication frequencies. Benefiting from these designs, the impedance bandwidth of the TCDA reaches 26.2:1 from 0.20 GHz to 5.23 GHz (VSWR < 3).

## 3. Finite Array

Then, TCDA elements are composed of a 6 × 6 finite array and the simulated boundary conditions are replaced by radiation boundary.

### 3.1. Simulation of Scanning Characteristics

In applications, TCDA often needs to be installed on fast-moving platforms such as cars, speedboats, and aircraft or needs to communicate with such targets. At this time, it is difficult to align with mechanical rotation structures alone. Therefore, it is meaningful to use a phased array to achieve beam scanning.

By adjusting the phase of the units’ feed ports of different rows, we can explore the beam scanning potential of TCDA. Figure 4 shows that the Active VSWR of the center element is less than 4 at most frequency points when scanning to 45° in both the E-plane and the H-plane. The simulation results show the designed TCDA can exhibit certain scanning characteristics if needed. This means that it can radiate directionally in the specified direction, which meets the above usage requirements.

### 3.2. Loading of Feed Network

However, such an array combination would mean 72 feed ports, making it too cumbersome in practical applications. To solve this problem, this paper innovatively proposes a highly integrated feed network. As shown in Figure 5, two 1 to 16 power dividers are designed to differential feed 32 ports of a 4 × 4 array. A 0.508 mm-thick RO4003 substrate is directly attached to the ground with a microstrip circuit printed on its bottom surface. The inner conductors of the twin coaxial lines are connected to the microstrip through the substrate and the outer conductors are in direct contact with the ground. WAIM, RFSS and ground are extended to obtain a better radiation effect making the overall size 236.4 mm × 236.4 mm × 74.4 mm, whose profile height is strictly limited at only 0.673 *λ_c_* and 0.05 *λ_low_*.

By adjusting the parameters such as the length and width of each microstrip line to achieve in-band impedance matching, an ultra-wideband feed network is designed. As is shown in Figure 6, the insertion loss of the feed network is less than −2 dB in most frequency bands, with an average of −1.02 dB. This indicates that it can replace the complex distributed feed mode with a feed with an acceptable loss. More details about the feed network lines can be seen in Figure A3.

Eventually, the radiation efficiency of the TCDA is depicted in Figure 7. Even if there are losses caused by WAIM and RFSS, the whole TCDA still has a maximum radiation efficiency of nearly 100% and an average value of 74.76%. It is significantly lower at 1.6 GHz and 4.4 GHz, mainly because the RFSS absorbs part of the electromagnetic wave. Moreover, the simulated realized gain reaches a maximum value of 17.98 dBi at 5 GHz and 7.85 dBi at 2.4 GHz. Therefore, it is feasible to use the feed network to reduce the complexity of TCDA under the premise of ensuring radiation characteristics.

## 4. Measured Results

To verify the actual engineering characteristics of the simulated TCDA, a 4 × 4 array is manufactured and tested in an anechoic chamber. Benefiting from the dedicated feed network design, the number of feed ports was reduced from 32 to only 2. Unlike when using phase-shifted transmission lines [22], this paper’s specific method is to divide the signal from the source into two equal amplitude and phase signals through a power divider. Then, connect four phase shifters in series to one of the signals. By respectively adjusting the phase shifters at each test frequency point, the phase differences between the two signals are moved to 180°.

The experimental device is shown in Figure 8. The TCDA is placed on a platform from 0.24 m to 2.4 m away from the probe, which is equivalent to four times the wavelength of the measured frequency. When it is being fed, the probe is moved to measure the voltage level in each direction. Gain can be obtained by comparing the level values of the TCDA and the standard rectangular horn antenna, which is detailed in the table of the submitted dataset. Thanks to the unique feed network, there is almost no need for much additional equipment other than TCDA itself during the experiment, showing great convenience. The profile height is only 74.5 mm, which is consistent with the designed value.

Figure 9 shows the comparison of measured and simulated VSWR of the TCDA, which is experienced by the Agilent E8363C network analyzer. Due to the influence of the feed port setting, machining accuracy and experimental error, VSWR is partly different from the simulated array. But it is still less than 3.3 in the whole frequency band and the fluctuation trend is roughly the same. The frequency points of 1.2 GHz and 1.6 GHz with the largest VSWR are caused by the loss of the feed network, which corresponds to Figure 6. In applications, such a wide working bandwidth can bring a great transmission rate.

Measured realized gain and radiation pattern are compared with the simulations in Figure 10 and Figure 11. As can be seen, the experimental data agree well with the finite array simulated data. The measured gain reaches 6.52 dBi and 17.19 dBi at the main working frequencies of 2.4 GHz and 5 GHz with merely 16 elements, which is close to the simulation. The test results also show that the side lobe levels of the E-plane and the H-plane at 5 GHz are −16.42 dB and −17.05 dB, and the half-power beam widths are 17° and 23.1°. These parameters are, respectively, −12.90 dB, −10.83 dB, 23.9°, and 29.1° at 4 GHz.

Due to the long wavelength at low frequencies, there is a certain deviation compared to the simulation of the ideal radiation boundary during experiments in the anechoic chamber. This is mainly because the longer wavelengths are closer to the size of the object, which interferes with the radiation pattern. However, the experimental and simulation results are basically consistent at high frequencies with shorter wavelengths. In general, the designed TCDA realizes high gain and great directivity with a small number of units, showing excellent radiation performance.

The error analysis includes the following three aspects:

Firstly, the production of the RFSS uses the carbon oil printing process to achieve custom resistance, which is adjusted manually. The error of the realized resistance value will affect the passage of electromagnetic waves.

Secondly, the manually adjusted phase shifters have deviations during the test, which is different from the standard 180° differential signal set in the simulation. As a result, the dipoles are not strictly differentially fed. This will have a certain impact on its radiation.

Lastly, other processing will also produce a certain impact, such as the error of precision structures like microstrip lines and slots of tightly coupled antennas, the impedance change caused by the solder joints, screws added to fix the structure, and so on.

All of these will result in differences between the test results and the simulation results in terms of VSWR, gain, and radiation pattern.

Table 1 compares this work and other TCDAs to highlight the advantages of the proposed antenna array. It can be seen that under the premise of containing the feed network and using only 16 elements, the designed TCDA of this paper achieves outstanding gain, bandwidth and low profile. Although some pieces of literature have achieved better parameters, their in-band minimum radiation efficiencies are all as low as 30–35% [8,9,12,23]. This indicates that compared with other works, the designed TCDA has outstanding indices, minimum size and concise feeding method, showing significant advantages.

## 5. Conclusions

Based on the proposed novel tightly coupled dipole structure and dual-stopband resistive frequency selective surface, a TCDA with an impedance bandwidth of 26.2:1 (0.20–5.23 GHz) is designed in this paper. It exhibits excellent radiation performance with an average radiation efficiency of 74.76% and a maximum measured gain of 17.19 dBi. At the same time, it benefits from the use of a high dielectric constant superstrate and the design of the feed network and antenna’s common ground, the antenna array achieves an extremely low profile of 0.05 *λ_low_*. The number of feed ports was also reduced to only two. The measured results are basically consistent with the simulation, which proves the feasibility of the designed antenna.

Therefore, the designed TCDA has broad application prospects in lightweight platforms like drones, cars, and other equipment with strict spatial limitations, which is of great significance in the context of the increasing importance of UAVs in the world. For example, drones with designed broadband high-rate antennas have important application value in emergency communication scenarios such as camping, disaster relief, and power outages. They can use ultra-wideband to communicate at a high rate or provide 2.4 and 5 GHz WIFI signal communication, demonstrating flexibility, fast responsiveness, high convenience and powerful communication capabilities. The challenge of communication in these scenarios is solved through their miniaturized structure and great radiation characteristics.

## Figures and Tables

**Figure 1 sensors-25-03418-f001:**
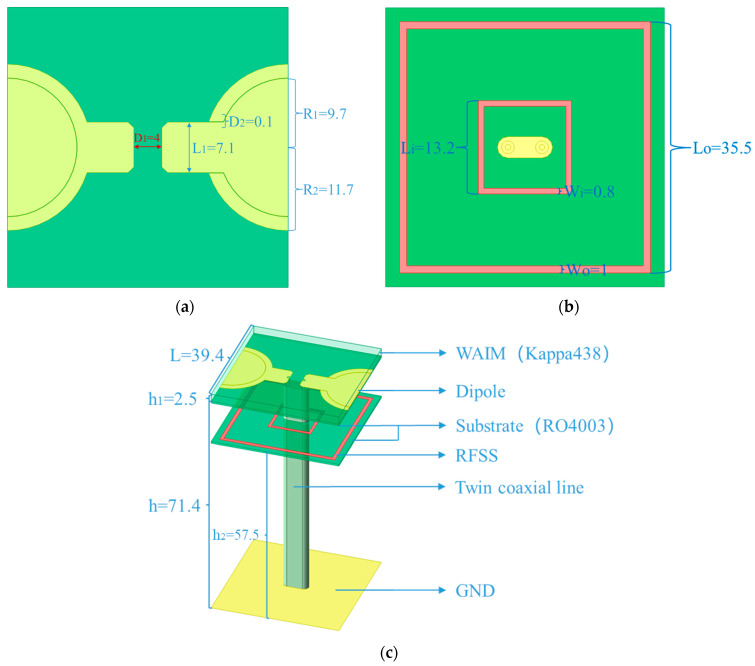
Unit cell of a TCDA with RFSS and WAIM (in mm): (**a**) new circular arc antenna structure, (**b**) dual-stopband RFSS, (**c**) complete structure.

**Figure 2 sensors-25-03418-f002:**
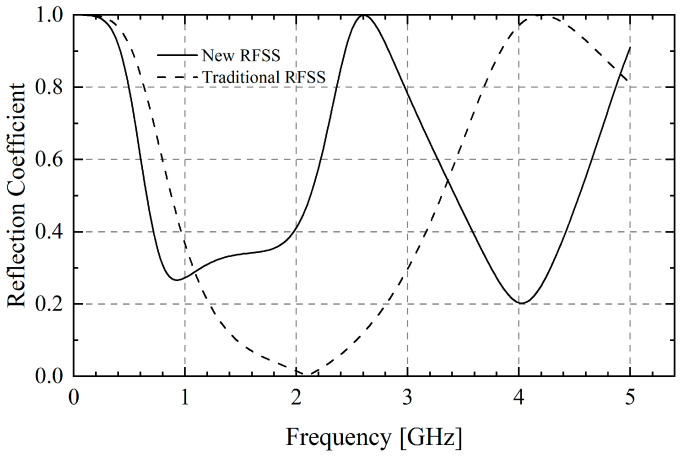
Comparison of reflection coefficient between new and traditional RFSS.

**Figure 3 sensors-25-03418-f003:**
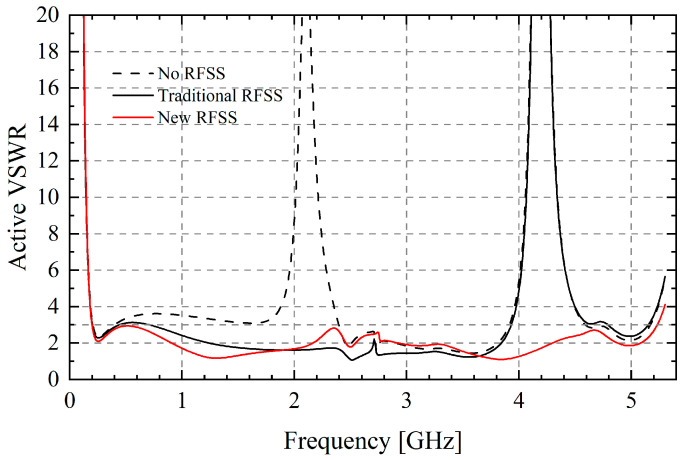
Active VSWR of infinite array and comparison between no RFSS and traditional RFSS.

**Figure 4 sensors-25-03418-f004:**
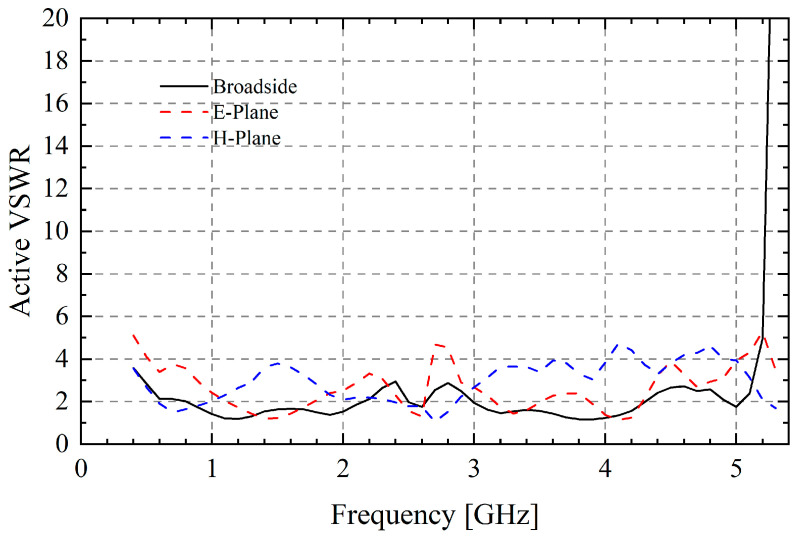
Active VSWR of finite TCDA when scanning to 45°.

**Figure 5 sensors-25-03418-f005:**
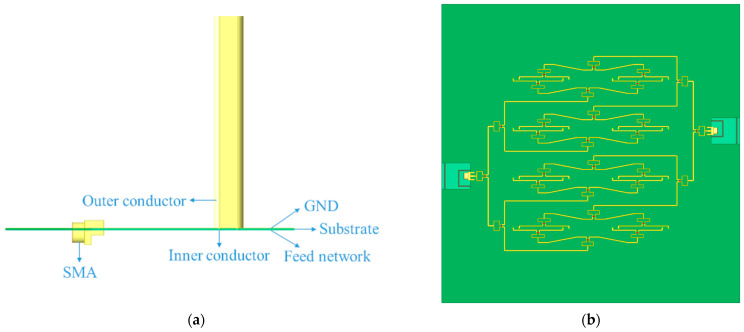
Structure of designed feed network: (**a**) front view, (**b**) bottom view. The substrate is cut with two gaps to place the SMA connector to make the microstrip line as short as possible, thereby reducing the loss.

**Figure 6 sensors-25-03418-f006:**
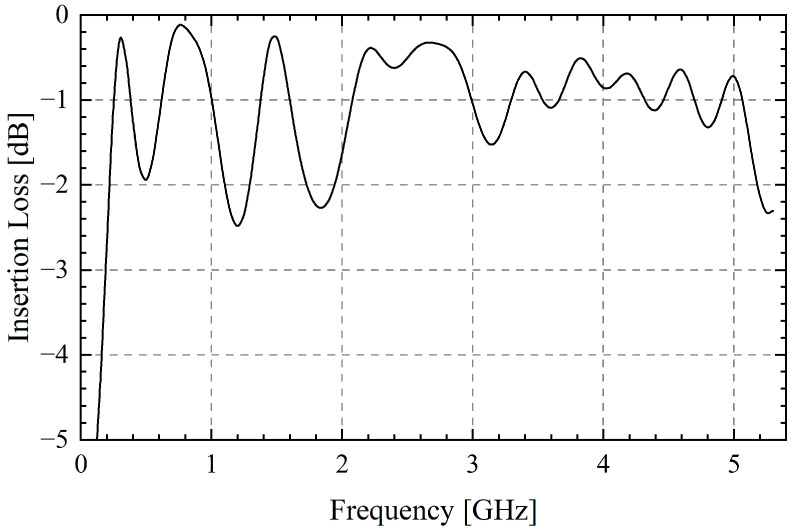
Insertion loss of feed network.

**Figure 7 sensors-25-03418-f007:**
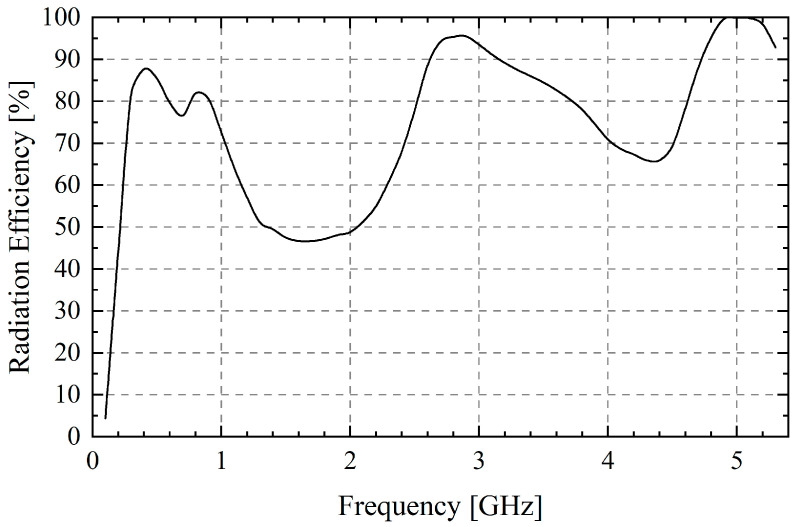
Radiation efficiency of TCDA.

**Figure 8 sensors-25-03418-f008:**
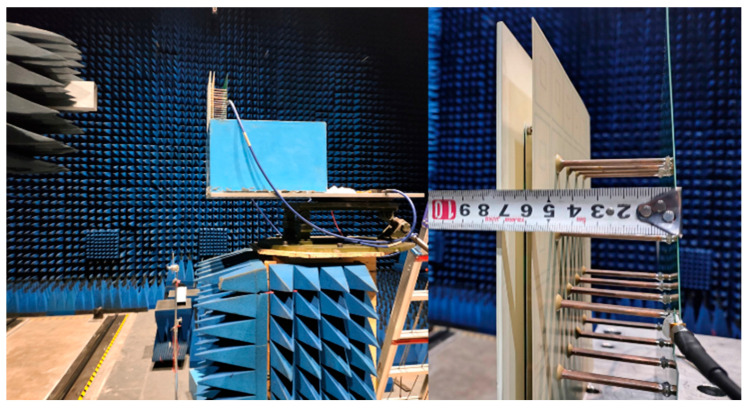
Tested 4 × 4 array prototype. Only two differential cables are needed to feed the TCDA. And profile height is exactly equal to 7.4 cm.

**Figure 9 sensors-25-03418-f009:**
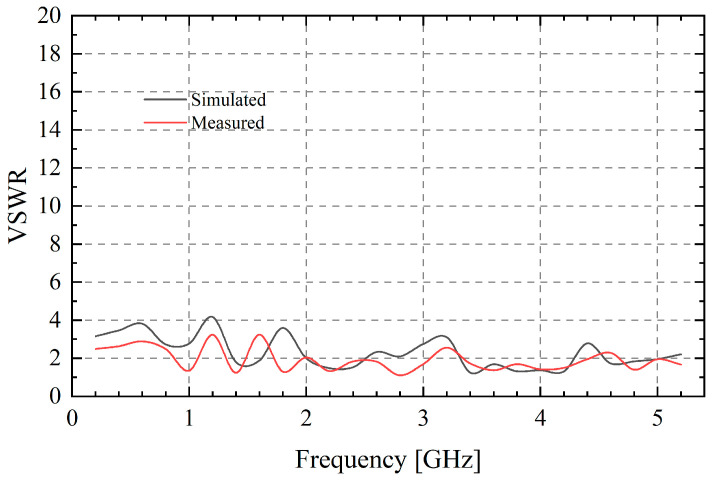
Simulated and measured VSWR performance of the proposed TCDA.

**Figure 10 sensors-25-03418-f010:**
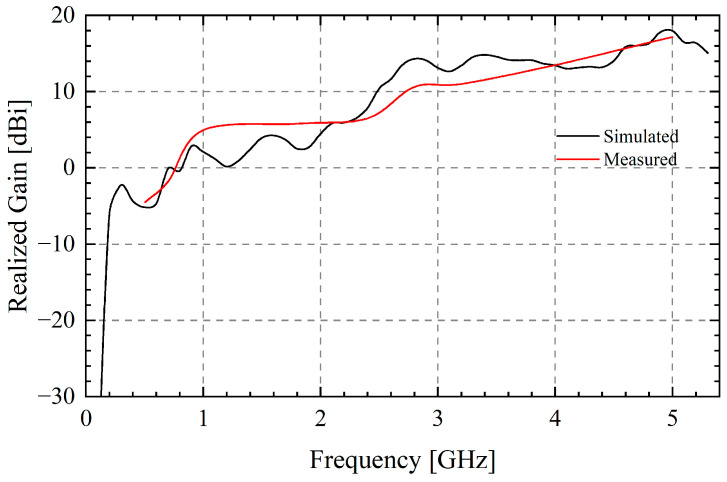
Simulated and measured realized gain.

**Figure 11 sensors-25-03418-f011:**
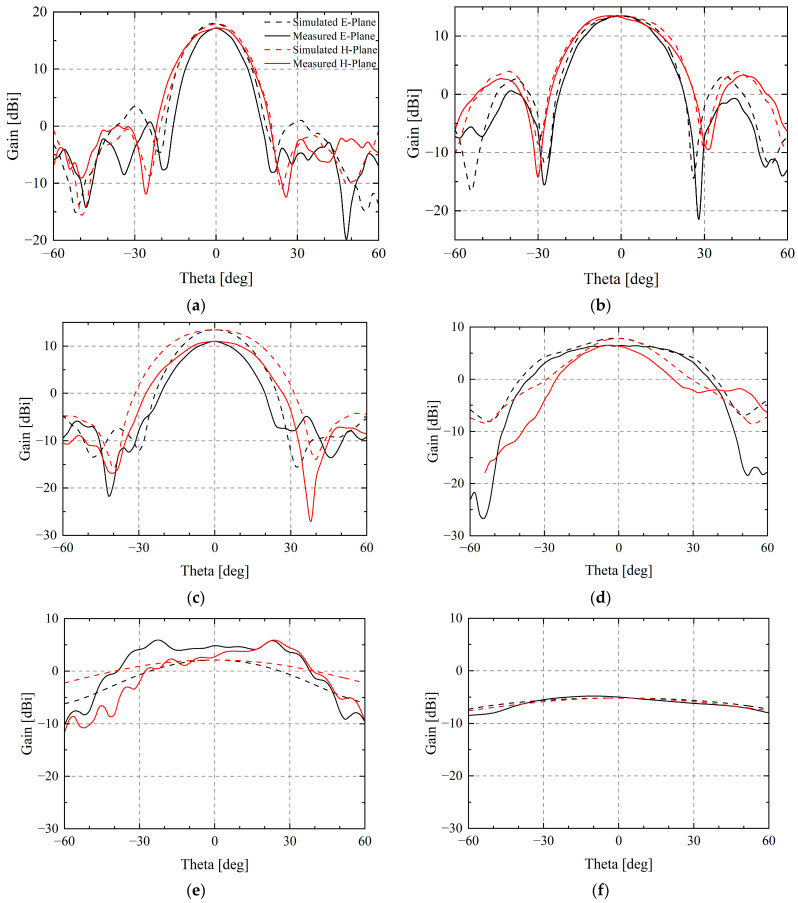
Radiation pattern of (**a**) 5 GHz, (**b**) 4 GHz, (**c**) 3.2 GHz, (**d**) 2.4 GHz, (**e**) 1 GHz, (**f**) 0.5 GHz. The solid lines indicate the measured results and the black lines indicate the E-plane. For frequencies ranging from low to high, the directional pattern tends from flat to directional.

**Table 1 sensors-25-03418-t001:** Comparison of TCDA performance.

Design	Bandwidth (GHz)	Number of Elements	Maximum Gain (dBi)	Min Radiation Efficiency (%)	Profile
[5]	21:1 (0.28–5.91)	4 × 4	16	73	0.055 *λ_low_*
[6]	10:1 (0.2–2.0)	6 × 10	15	81	0.064 *λ_low_*
[8]	46.1:1 (0.13–6)	12 × 12	15	30	0.074 *λ_low_*
[9]	24:1(1–24)	8 × 8	23	32	0.04 *λ_low_*
[11]	10.5:1 (0.86–9)	10 × 10	21	\	0.135 *λ_low_*
[12]	28:1 (0.20–5.6)	12 × 12	13	35	0.069 *λ_low_*
[23]	50:1 (0.33–16.5)	10 × 10	23	34	0.05 *λ_low_*
[24]	23.2:1 (0.24–5.1)	16 × 16	22	45	0.073 *λ_low_*
This work	26.2:1 (0.20–5.23)	4 × 4	17.19	47	0.05 *λ_low_*

## Data Availability

The dataset can be found at DOI:10.21227/084h-d533.

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
