# Peer review of "A 26.2:1 Bandwidth Ultra-Wideband Low-Profile Tightly Coupled Dipole Array with Integrated Feed Network"

_sensors, 2025, doi:10.3390/s25113418_

Round 1
Reviewer 1 Report
Comments and Suggestions for Authors
The paper shows some solid improvements in bandwidth, gain, and how compact the antenna array is compared to older designs. The part about applications is interesting, but it feels a bit light on technical depth. It would help if they could give some numbers—like estimated data speeds or how well the signal holds up in real-world drone emergency setups.
A quick note on beam scanning would be useful too, since drones move around a lot and that affects performance. The measurement section could use more detail, how far apart the test antenna and signal source were, and what equipment they ran the tests with.
Figure 9 only shows measured results, but it’d be better to see simulations side by side for comparison. And for the gain values, it’d be good to know why measurement didn’t quite match the simulations.
Comments on the Quality of English LanguageIt can be improved
Author Response
Comments1: The part about applications is interesting, but it feels a bit light on technical depth. It would help if they could give some numbers—like estimated data speeds or how well the signal holds up in real-world drone emergency setups.
Response1: We have added the calculation method for the maximum transmission rate of TCDA theory on page 8. Two emergency signal provision methods and their advantages were also added to the conclusion.
Comments2: A quick note on beam scanning would be useful too, since drones move around a lot and that affects performance.
Response2: We added a description and summary of its application scenarios based on the reviewer's comment in section 3.1 to demonstrate its value and increase readers' interest.
Comments3: The measurement section could use more detail, how far apart the test antenna and signal source were, and what equipment they ran the tests with.
Response3: We have added information on the distance between the antenna and the source, the equipment used, and the gain calculation method on page7 based on the details mentioned in the comment. This will help readers better understand the experimental methods.
Comments4: Figure 9 only shows measured results, but it’d be better to see simulations side by side for comparison.
Response4: We previously believed that the VSWR measured in the experiment was from a single source, while there were two ports in the simulation. So the experimental results could not be directly compared with the simulation results and we did not plot the simulation results in Figure 9. After considering the comment of the reviewers, we plotted the average VSWR of the two simulated ports in Figure 9 and compared them. Although there are some differences in amplitude and frequency corresponding to extreme values between them, the overall trend of change is the same.
Comments5: And for the gain values, it’d be good to know why measurement didn’t quite match the simulations.
Response5: On page 9, we analyzed the reasons of error. After considering the comment of the reviewer, in order to better explain the differences in radiation images, we have supplemented the impact of wavelength on experimental accuracy on pages 8-9. In addition, we also added the impact of the screws used in the processing on page 10.
Reviewer 2 Report
Comments and Suggestions for Authors The paper presents a 26.2:1 bandwidth ultrawideband, low-profile tightly coupled dipole array. My comments are as follows: 1. The authors claim that their novel RFSS can absorb in multiple bands. However, prior work has already demonstrated similar capabilities. For example: ref [6], [R1], [R2], [R3]. It would strengthen the manuscript if the authors could more clearly articulate how their approach is novel compared to existing literature and elaborate on the specific contributions of their design. [R1] Y. Feng, Y. Chen and M. Zhu, "A 50:1 Bandwidth Tightly Coupled Dipole Array With Short-Circuit Resonance Suppressing Surface," in IEEE Transactions on Circuits and Systems II: Express Briefs, vol. 71, no. 7, pp. 3323-3327, July 2024.[R2] R. Solanki, P. Khiang Tan and T. Huat Gan, "Extending the Low Frequency Limit of Tightly Coupled Dipole Arrays for Ultra-Wide Bandwidth With Ultra-Thin Profile," in IEEE Access, vol. 12, pp. 184929-184939, 2024. [R3] M. W. Nichols, M. O. Anastasiadis, M. E. Taaffe, E. A. Alwan, and J. L. Volakis, ‘‘Ultra-wideband tightly coupled dipole array fed by a tapering meandered balun,’’ IEEE Open J. Antennas Propag., vol. 4, pp. 936–946, 2023. 2. The literature review could be improved. For example: [R1] demonstrates a 50:1 bandwidth, which surpasses that of the proposed design. [R2] achieves a similar bandwidth (24:1) with a much thinner profile (approximately 0.04λL). [R3] reports a 42:1 bandwidth, again exceeding the proposed work’s performance. Several other studies with superior or comparable results have been overlooked. A more comprehensive and critical comparison with state-of-the-art designs is necessary to contextualize the significance of the proposed work. 3. The measured gain results show a noticeable discrepancy from the simulated results. The authors are encouraged to provide an explanation for this deviation. Furthermore, the manuscript currently lacks measured gain and radiation pattern below 1 GHz, which is important for substantiating the performance claims in the lower end of the band.
Author Response
Comments1: The paper presents a 26.2:1 bandwidth ultrawideband, low-profile tightly coupled dipole array. My comments are as follows: The authors claim that their novel RFSS can absorb in multiple bands. However, prior work has already demonstrated similar capabilities. For example: ref [6], [R1], [R2], [R3]. It would strengthen the manuscript if the authors could more clearly articulate how their approach is novel compared to existing literature and elaborate on the specific contributions of their design.
[R1] Y. Feng, Y. Chen and M. Zhu, "A 50:1 Bandwidth Tightly Coupled Dipole Array With Short-Circuit Resonance Suppressing Surface," in IEEE Transactions on Circuits and Systems II: Express Briefs, vol. 71, no. 7, pp. 3323-3327, July 2024.
[R2] R. Solanki, P. Khiang Tan and T. Huat Gan, "Extending the Low Frequency Limit of Tightly Coupled Dipole Arrays for Ultra-Wide Bandwidth With Ultra-Thin Profile," in IEEE Access, vol. 12, pp. 184929-184939, 2024.
[R3] M. W. Nichols, M. O. Anastasiadis, M. E. Taaffe, E. A. Alwan, and J. L. Volakis, ‘‘Ultra-wideband tightly coupled dipole array fed by a tapering meandered balun,’’ IEEE Open J. Antennas Propag., vol. 4, pp. 936–946, 2023.
Response1: Thanks very much for the reviewer's work and assistance in the literature review. We have added all the references mentioned in the comments to the article. For convenience, the reference numbers [6], [R1], [R2], [R3] mentioned in the comments have been replaced with the reference numbers [7], [23], [9], [10] from the new submission.
We cited references [7-10] in the description of the literature on implementing multi stopband functionality in previous work on page 2. But for reference [23], we confirmed in its description on the fourth page that its FSS can only absorb electromagnetic waves at 4GHz, so we did not cite it here. To demonstrate the differences between our work and them, we explained on pages 2-4 that they all use multi-layer or complex serrated structures. But this article uses two rings corresponding to the resonant frequencies, which have the simplest form that can theoretically exist and can greatly reduce the difficulty of design and processing.
Comments2: The literature review could be improved. For example: [R1] demonstrates a 50:1 bandwidth, which surpasses that of the proposed design. [R2] achieves a similar bandwidth (24:1) with a much thinner profile (approximately 0.04λL). [R3] reports a 42:1 bandwidth, again exceeding the proposed work’s performance. Several other studies with superior or comparable results have been overlooked. A more comprehensive and critical comparison with state-of-the-art designs is necessary to contextualize the significance of the proposed work.
Response2: We have added a comparison of references [9] and [23] in Table 1. However, for reference [10], the key parameter of gain was not provided, making it unable to make a comparison. At the same time, there is already a literature [8] with similar bandwidth in the table, so it was not added.
We found that for all works with better bandwidth or profile height than in this article, the lowest in-band radiation efficiency is very low. Therefore, we also added a comparison of this item in Table 1 and explained it on page 10. This once again demonstrates the significant indicator advantages of the TCDA proposed in this article.
Comments3: The measured gain results show a noticeable discrepancy from the simulated results. The authors are encouraged to provide an explanation for this deviation.
Response3: On page 9, we analyzed the reasons of error. After considering the comment of the reviewer, in order to better explain the differences in radiation images, we have supplemented the impact of wavelength on experimental accuracy on pages 8-9. In addition, we also added the impact of the screws used in the processing on page 10.
Comments4: The manuscript currently lacks measured gain and radiation pattern below 1 GHz, which is important for substantiating the performance claims in the lower end of the band.
Response4: We have added a 0.5GHz radiation pattern in Figure 11 to characterize the situation below 1GHz, which is the frequency corresponding to the largest standard rectangular horn that can be found in this region, as the wavelength has already reached 0.6m. In addition, we have also added a 3.2GHz pattern to help readers to have a more continuous understanding of the radiation situation of TCDA.
Round 2
Reviewer 2 Report
Comments and Suggestions for Authors
My previous review comments have been addressed. I have only one additional comment:
In the newly added paragraph on page 8, the channel capacity calculation is not necessary, as bandwidth and SNR vary depending on the specific application. Moreover, there is a typographical error in the calculation—20 dB corresponds to an SNR of 100, whereas the equation uses an SNR of 1000.
Author Response
Comments1:In the newly added paragraph on page 8, the channel capacity calculation is not necessary, as bandwidth and SNR vary depending on the specific application. Moreover, there is a typographical error in the calculation—20 dB corresponds to an SNR of 100, whereas the equation uses an SNR of 1000.
Response1:Thank you for your prompt and patient review. We have removed the calculations and values regarding specific transmission rate from page 8 and the conclusion. Only the description of high rate is retained for greater rigor.